# Organization of DNA Replication Origin Firing in *Xenopus* Egg Extracts: The Role of Intra-S Checkpoint

**DOI:** 10.3390/genes12081224

**Published:** 2021-08-09

**Authors:** Diletta Ciardo, Olivier Haccard, Hemalatha Narassimprakash, Jean-Michel Arbona, Olivier Hyrien, Benjamin Audit, Kathrin Marheineke, Arach Goldar

**Affiliations:** 1Institute of Integrative Biology of the Cell (I2BC), CNRS, CEA, University Paris Sud, 1, Avenue de la Terrasse, 91190 Gif-sur-Yvette, France; ciardo@bio.ens.psl.eu (D.C.); olivier.haccard@i2bc.paris-saclay.fr (O.H.); hemalatha.narassimprakash@i2bc.paris-saclay.fr (H.N.); kathrin.marheineke@i2bc.paris-saclay.fr (K.M.); 2University Lyon, ENS de Lyon, Laboratoire de Biologie et Modélisation de la Cellule, University Claude Bernard Lyon 1, CNRS UMR5239, INSERM U1210, 46 Allé d’Italie Site Jacques Monod, 69007 Lyon, France; jeanmichel.arbona@ens-lyon.fr; 3Institut de Biologie de l’Ecole Normale Supérieure (IBENS), Ecole Normale Supérieure, CNRS, INSERM, PSL Research University, 75005 Paris, France; hyrien@biologie.ens.fr; 4University Lyon, ENS de Lyon, University Claude Bernard Lyon 1, CNRS, Laboratoire de Physique, 69342 Lyon, France; benjamin.audit@ens-lyon.fr

**Keywords:** DNA replication, checkpoint, mathematical modelling, minimal model, origin firing

## Abstract

During cell division, the duplication of the genome starts at multiple positions called replication origins. Origin firing requires the interaction of rate-limiting factors with potential origins during the S(ynthesis)-phase of the cell cycle. Origins fire as synchronous clusters which is proposed to be regulated by the intra-S checkpoint. By modelling the unchallenged, the checkpoint-inhibited and the checkpoint protein Chk1 over-expressed replication pattern of single DNA molecules from *Xenopus* sperm chromatin replicated in egg extracts, we demonstrate that the quantitative modelling of data requires: (1) a segmentation of the genome into regions of low and high probability of origin firing; (2) that regions with high probability of origin firing escape intra-S checkpoint regulation and (3) the variability of the rate of DNA synthesis close to replication forks is a necessary ingredient that should be taken in to account in order to describe the dynamic of replication origin firing. This model implies that the observed origin clustering emerges from the apparent synchrony of origin firing in regions with high probability of origin firing and challenge the assumption that the intra-S checkpoint is the main regulator of origin clustering.

## 1. Introduction

Eukaryotic genomes are duplicated in a limited time during the S phase of each cell cycle. Replication starts at multiple origins that are activated (fired) at different times in S phase to establish two diverging replication forks that progress along and duplicate the DNA at fairly constant speed until they meet with converging forks originated from flanking origins [1,2]. The mechanisms that regulate the timing of origin firing remain largely unknown [3,4,5,6,7,8].

The core motor component of the replicative helicase, the MCM2-7 complex, is loaded on chromatin from late mitosis until the end of G1 phase as an inactive head-to-head double hexamer (DH) to form a large excess of potential origins [9,10]. During S phase, only a fraction of the MCM2-7 DHs are activated to form a pair of active Cdc45-MCM2-7-GINS (CMG) helicases and establish bidirectional replisomes [1]. MCM2-7 DHs that fail to fire are inactivated by forks emanating from neighboring fired origins [11]. Origin firing requires S-phase cyclin-dependent kinase (CDK) and Dbf4-dependent kinase (DDK) activities as well as the CDK targets Sld2 and Sld3 and the replisome-maturation scaffolds Dpb11 and Sld7 in *S. cerevisiae*. The six initiation factors Sld2 (RecQ4 in *Xenopus*), Sld3 (Tresline in *Xenopus*), Dpb11 (TopBP1 in *Xenopus*), Dbf4 (Drf1 in *Xenopus*), Sld7 (MTBP in *Xenopus*) and Cdc45 are expressed at concentrations significantly lower than the MCM complex and core replisome components, suggesting that they may be rate-limiting for origin firing [12,13]. Among these six factors, Cdc45 is the only one to travel with the replication fork.

DNA replication initiates without sequence specificity in *Xenopus* eggs [14,15], egg extracts [16,17,18,19] and early embryos [20,21] (for review, see [22]). To understand how a lack of preferred sequences for replication initiation is compatible with a precise S-phase completion time, investigators have studied replication at the single DNA molecule level using the DNA combing technique [23,24,25,26,27] which contrast to population based approaches, that average replication characteristics. DNA combing technique reveals cell-to-cell differences in origin activation important for understanding how genomes are replicated during S-phase, these experiments did not detect a regular spacing of initiation events but revealed that the origin firing rate strongly increases from early to late replication intermediates, speeding up late replication stages [23,24]. An observation that has been also confirmed in many other model organisms, including human cell lines [28].

A mathematical model based on the assumptions (i) that the probability of firing of each replication origin can be replaced by the averaged probability of firing calculated over all degree of freedom of origin firing process (MCM2-7 DH density, genomic position, chromatin compaction, nucleosome density, etc. named “mean-field hypothesis”), (ii) that firing of origins are independent events and (iii) that fork speed is constant was proposed [29]. This model allowed the extraction of a time-dependent rate of replication initiation, $I\left(t\right)$, from the measured eye lengths, gap lengths and eye-to-eye distances on combed DNA molecules (Figure 1a) [29]. The extracted $I\left(t\right)$ markedly increased during S phase. Simulations incorporating this extracted $I\left(t\right)$ reproduced the mean eye length, gap length and eye-to-eye distance, but the experimental eye-to-eye distance distribution appeared “peakier” than the simulated one [22,30]. Modulating origin firing propensity by the probability to form loops between forks and nearby potential origins resulted in a better fit to the data without affecting $I\left(t\right)$ [30].

Importantly, experiments revealed that in *Xenopus*, like in other eukaryotes, replication eyes are not homogeneously distributed over the genome but tend to cluster [25,27]. First, a weak correlation between the sizes of neighbouring eyes was observed [25,27,30], consistent with firing time correlations. Second, more molecules with no or multiple eyes than expected for spatially uniform initiations were observed in replicating DNA [27]. There are two potential, non-exclusive mechanisms for these spatiotemporal correlations. The first one, compatible with a mean-field hypothesis, is that activation of an origin stimulates nearby origins. The second one, no longer consistent with a mean-field hypothesis, is that the genome is segmented into multi-origin domains that replicate at different times in S phase. This second hypothesis has been explored numerically in human and has been shown to be compatible with the universal bell shaped $I\left(t\right)$ profile [31].

Interestingly, experiments in *Xenopus* egg extracts revealed that intranuclear replication foci labelled early in one S phase colocalized with those labelled early in the next S phase, whereas the two labels did not coincide at the level of origins or origin clusters were examined [32]. Given the different characteristic sizes of timing domains (1–5 Mb) and origin clusters (50–100 kb) in the *Xenopus* system, it is possible that origin correlations reflect both a programmed replication timing of large domains and a more local origin cross-talk within domains.

It is now well accepted that the intra-S phase checkpoint regulates origin firing during both unperturbed and artificially perturbed S phase [27,33,34,35,36]. DNA replication stress, through the activation of the S-phase checkpoint kinase Rad53, can inhibit origin firing by phosphorylating and inhibiting Sld3 and Dbf4 [37]. The metazoan functional analogue of Rad53 is Chk1. Experiments in human cells under low replication stress conditions showed that Chk1 inhibits the activation of new replication factories while allowing origin firing to continue within active factories [33]. Experiments using *Xenopus* egg extracts suggested that the checkpoint mainly adjusts the rate of DNA synthesis by staggering the firing time of origin clusters [27]. Our first model for DNA replication in *Xenopus* egg extracts [38] (which combined time-dependent changes in the availability of a limiting replication factor, and a fork-density dependent affinity of this factor for potential origins) was used to model the regulation of DNA replication by the intra-S checkpoint [35]. We showed that even during an unperturbed S phase in *Xenopus* egg extracts, Chk1 inhibits origin firing away from but not near active forks [35]. To account for the regulation of DNA replication by the intra-S checkpoint, we replaced the dependency of origin firing on fork density by a Chk1-dependent global inhibition of origin firing with local attenuation close to active forks as was proposed in other contexts [33,39,40,41]. This model was able to simultaneously fit the $I\left(f\right)$ (the rate of origin firing expressed as a function of each molecule’s replicated fraction *f*) of both a control and a UCN-01-inhibited Chk1 replication experiment [35]. However, in that work, we did not push further the analysis to verify if our model was able to explain simultaneously $I\left(f\right)$ (temporal program) and the eye-to-eye distance distribution (spatial program).

In the present work, using numerical simulations, we quantitatively analysed both the temporal and spatial characteristics of genome replication as measured by DNA combing in the in vitro *Xenopus* system. *Xenopus* egg extracts have been successfully used since over three decades now to study DNA replication in metazoans [42]. Rooted on experimental data, we build a general and minimal model of DNA replication able to predict both the temporal and the spatial characteristics either during an unchallenged or a challenged S phase. We use the experimental data from [35] where the experimental mean chosen for activating or inhibiting (manipulating) the checkpoint was respectively to overexpress Chk1 protein or to inhibit its activity using the specific inhibitor UCN-01. By analysing the spatio-temporal pattern of DNA replication after inhibition or activation of intra-S checkpoint and comparing them to an unchallenged pattern we disentangle the complex role of the intra-S checkpoint for replication origin firing.

## 2. Materials and Methods

### 2.1. Replication of Sperm Nuclei in *Xenopus* Egg Extracts

DNA combing data using the Xenopus in vitro system from Platel et al. 2015 [35] were analysed. Briefly, sperm nuclei (2000 nuclei/μL) were incubated in *Xenopus* egg extracts in the presence of cycloheximide, energy mix and 20 μM biotin-dUTP (Roche Applied Science). UCN-01 (Selleck Chemicals) (or solvent (DMSO) alone as control) was added at 1 μM. Replication was allowed to continue for 40, 60 or 75 min. In order to increase the number of eye-to-eye distances in control samples at 40 min of the first experiment, data from two additional independent experiments with nearly identical replication content were combined for early S phase (45 and 50 min, respectively) (control 8%, UCN-01 22% replication.) For Chk1 overexpression experiments recombinant and active XChk1 with a N-terminal His-tag was expressed in the baculovirus expression system as described in Platel et al. 2015 [35]. We chose to moderately overexpress XChk1 threefold by adding 120 nM purified XChk1 (or dialysis buffer as control) in two independent replication reactions, which were stopped at 45 or 55 min, respectively. Labelled DNA was purified and DNA combing was performed as described in Platel et al., 2015 [35].

### 2.2. Monte Carlo Simulation of DNA Replication Process

A dynamical Monte Carlo method was used to simulate the DNA replication process as detailed before [38]. We simulate the replicating genome as a one-dimensional lattice of $L=10^{6}$ blocks of value 1 for replicated and 0 for unreplicated state, respectively. To match the spatial resolution of DNA combing experiments each block represents 1 kb. After one round of calculation an existing replication track grows in a symmetric manner by 2 blocks. Considering that the fork speed v=0.5kb min${}^{-1}$ is constant (except in MM2 where the value of *v* (kb min${}^{-1})$ for each active fork is randomly chosen at each round of calculus from {0,1,2,3} (kb min${}^{-1})$), one round of calculation corresponds to 2 min. In the continuous case we assume that the potential replication origins are continuously distributed on the genome with an average density of one potential origin per 1 kb (1 block). As it is also considered that potential replication origins are discrete objects and as a consequence are distributed in a heterogeneous manner on the genome [43,44] we also simulate the case where the distribution of potential origins is discrete. In the discrete case we assume that potential origins are randomly distributed along the genome with an average density of one potential origin per 2.3 kb [45]. In both cases origins fire stochastically. Origin firing requires an encounter with a trans-acting factor which number $N\left(t\right)$ increases as S phase progresses with a rate *J*, $N\left(t\right)=N_{0}+Jt$. If an encounter leads to an origin firing event, the trans-acting factor is sequestrated by replication forks and hence the number of available trans-acting factors is $N_{f}\left(t\right)=N\left(t\right)-N_{b}\left(t\right)$, where $N_{b}\left(t\right)$ is the number of bound factors. To ensure that origins do not re-fire during one cycle and are inactivated upon passive replication, only “0” blocks (not replicated) are able to fire. Before the beginning of replication process the one-dimensional lattice is randomly segmented into θL blocks where the probability of origin firing is $P_{in}$ and $\left(1-\theta \right)L$ blocks where the probability of origin firing is $P_{out}$. After the start of replication process, at each round of calculus, each block is randomly assigned a value between 0 and 1. This value is compared to $P_{in}$ or $P_{out}$ (depending to which category the block belongs) to decide whether the block may fire. In total, *M* “0” blocks (M≤L) with value strictly smaller than their reference probability may fire. If $M\;\leq \;N_{f}\left(t\right)$ all *M* blocks may fire, otherwise $N_{f}\left(t\right)$ blocks may fire. Furthermore, in MM3 and MM5, we consider that the probability of origin firing $P_{local}$ may be increased downstream of a replication fork over a distance *d*. The trans-acting factors sequestered by forks are released and are made available for new initiation events when forks meet.

### 2.3. Measuring: The Replicated Fraction $f\left(t\right)$, the Rate of Origin Firing $I\left(t\right)$, Fork Density $N_{fork}\left(t\right)$, Eye-to-Eye, Eye and Gap Length Distributions

The genome is represented as a one-dimensional lattice of $10^{6}$ elements $x_{i}\in \left\{0,1\right\}$. At each round of calculation the replicated fraction is calculated as $f\left(t\right)={\langle x\rangle }_{i}$ corresponding to the average value of $x_{i}$ over the genome.

The rate of origin firing per length of unreplicated genome per time unit (3 min) is calculated at each round of calculation, by counting the number of newly created “1” blocks, $N_{1}$ and $I\left(t\right)=\frac{N_{1}}{\left(1-f\left(t\right)\right)L\Delta t}$ where Δt=3min and $L=10^{6}$. The density of replication forks is calculated at each round of calculation by counting the number of “01” tracks, $N_{left}$, and “10” tracks, $N_{right}$ and $N_{forks}\left(t\right)=\frac{N_{right}+N_{left}}{L}$. The distributions of eye-to-eye distances, eye lengths and unreplicated gap sizes are then computed from the distribution of “0” and “1” tracks after reshaping the data (see below).

### 2.4. Comparing Experimental and Numerical Data

The simulation results were compared to the DNA combing data from Platel et al. [35]. The fluorescence intensities for total DNA and replicated tracks of each fiber were measured and binarized on a Matlab ^®^ platform by using a thresholding algorithm. The threshold value was chosen to minimize the difference between the replicated fraction measured by $\alpha^{32}$P-dATP incorporation and by DNA combing. Replicated tracks larger than 1kb were scored as eyes. Gaps were considered significant if >1 kb, otherwise the two adjacent eyes were merged. The eyes whose lengths span from 1 to 3 kb were considered as new origin firing events. The time interval in which these new detectable events can occur was calculated as Δt=3 min assuming a constant replication fork velocity of v≈ 0.5 kb min${}^{-1}$. This data reshaping protocol was also applied to simulated DNA molecules, in order to match the spatial and temporal resolutions between the experimental and simulated data. The global replicated fraction of each sample was computed as the sum of all eye lengths divided by the sum of all molecule lengths. To minimize finite molecule length effects in comparisons between data and simulations, the experimental molecule length distribution was normalised and considered as probability density of molecule length in the sample and used to weight the random shredding of the simulated genome at each time (Figure 1b). The global replication fraction of simulated cut molecules was calculated. Only molecules from the simulation time that had the same global replication fraction as the experimental sample were further considered.

Molecules were sorted by replicated fraction $f\left(t\right)$. The rate of origin firing and fork density were calculated for each molecule as a function of $f\left(t\right)$ ($I\left(f\right)$ and $N_{fork}\left(f\right)$, respectively) for both simulated and experimental data. The experimental $I\left(f\right)$, $N_{fork}\left(f\right)$, eye-to-eye distances, eye and gap length distributions were computed as the averaged value of three independent experiments.

### 2.5. Modeling Experimental Data: Parameters Optimization

To estimate the parameters of the model, we fitted the six experimental observables ($I\left(f\right)$, $N_{fork}\left(f\right)$, replicated fibre, eye-to-eye distances, eye and gap length distribution) using a genetic optimization algorithm (Matlab ^®^). The fitness function was defined as the sum of the square of the differences between experimental and simulated data curves divided by the squared mean of the experimental data curve. The genetic optimization algorithm was set over three subpopulations of 20 individuals with a migration fraction of 0.1 and a migration interval of 5 steps. Each individual defined a set of variables for the simulation and the variables were chosen within the range reported in Table 1 for the model that best fit the data. At each generation, 3 elite children were selected for the next generation. The rest of the population corresponds to a mixture between 60% of children obtained after a scattered crossover between two individuals selected by roulette wheel selection and 40% of children obtained by uniform mutation with a probability of 0.2, leading to a variability of 8%. The genetic algorithm was stopped after 50 generations corresponding to the convergence of the optimization method. As the size of variable space is unknown, we considered a large domain of validity for the variables. This has as an effect to reduce the probability that the optimization process reaches a unique global minimum. For this reason, we repeat the genetic optimization method 100 times independently over each data set and consider for each optimization round only the best elite individual.

## 3. Results

### 3.1. Finding the Best Integrative Model of Unperturbed S Phase

Our previous model [35] failed to simultaneously reproduce the eye-to-eye distance distribution and the $I\left(f\right)$ of the same control experiment (Figure 2). This discrepancy could be explained if initiation events have a strong tendency to cluster [25,27]. Clustering produces an excess of small (intra-cluster) and large (inter-cluster) eye-to-eye distances compared to random initiations, but only the former could be detected on single DNA molecules due to their finite length [27]. Chk1 action has been proposed to regulate origins clusters [33]. However, Chk1 inhibition by UCN-01 did not result in the broader eye-to-eye distribution predicted by random origin firing (Figure 2c,d), suggesting that other mechanisms than intra-S checkpoint are involved in the origin clustering.

We therefore explored the ability of several nested models with growing complexity (designated MM1 to MM5) (Appendix A). MM1 corresponds to a mean field hypothesis of origin firing: all potential origins have a constant firing probability $P_{out}$ [38,46]. MM2 corresponds to MM1 but assuming that replication forks can have a variable speed [47,48]. MM3 corresponds to MM1 with a local perturbation, whereby the proximity of forks facilitates origin firing [30,49] over a distance *d* downstream of an active fork where the probability of origin firing is $P_{local}$. In MM4 origin firing does not follow the mean field hypothesis but assumes that the genome can be segmented into regions of high and low probabilities of origin firing [31,49] as accepted for most eukaryotes [8,43,50,51,52,53,54]. In this scenario, the probability of origin firing of potential origins located within a fraction θ of the genome, $P_{in}$, is assumed to be higher than the firing probability $P_{out}$ of potential origins in the complementary fraction 1−θ. Lastly, MM5 combines the specific features of MM3 and MM4 into a single model. Furthermore, to verify if the localized nature of potential origins [43,44] can influence the spatio-temporal program of origin firing, each considered scenario was simulated assuming either a continuous or a discrete distribution of potential origins except for MM2.

For each model, we coupled dynamic Monte Carlo numerical simulations to a genetic optimization algorithm to find the family of variables that maximized the similarity between the simulated and measured profiles of $I\left(f\right)$, replicated fraction of single molecules, global fork density, eye-to-eye distances, gap lengths and eye lengths. MM5 with localized potential origins (Figure 3) provided the best fit to the experimental data (Appendix A). The increase in concordance between MM5 and the data occurs at the expense of increasing the number of parameters, which is justifiable on statistical grounds (Appendix A) and the predictive ability of MM5 is verified (Appendix A).

We used MM5 to analyse unchallenged, checkpoint inhibited and Chk1 over expressed S phase (Appendix A). In all cases MM5 was able to model concomitantly $I\left(f\right)$ and eye-to-eye distance distribution (Figure 4). In conclusion, while MM5 does not include all the possible mechanisms involved in DNA replication process and its regulation, it can adequately predict the spatio-temporal dynamics of DNA replication and its regulation by checkpoint mechanisms using a limited number of processes.

### 3.2. Retrieving the Dynamics of an Unchallenged S Phase Using the MM5 Model

MM5 faithfully reproduced the temporal and spatial program of DNA replication from unperturbed S phase samples with global replicated fractions of 8%, 19% and 53% (Appendix A; Appendix A). The fitted values of parameters changed as S phase progressed (Figure 5).

However, only changes in *J*, θ, $P_{out}$ and *d* were statistically significant (Appendix A). In particular, we found that *J* increased from 8% to 19% replication and then dropped back at 53% replication. θ and $P_{out}$ increased only from 8% to 19% replication but not later, while *d* stayed constant between 8% and 19% replication and decreased at 53% replication.

These observations suggest that during an unchallenged S phase both the fraction (θ) of the genome with high probability of origin firing and the background probability ($P_{out}$) of origin firing outside that fraction increase as S phase progresses. Interestingly, $P_{local}$ is higher than $P_{in}$ and $P_{out}$, suggesting that firing of an potential origin significantly favours the firing of nearby potential origins over a distance *d*, compatible with a chromatin looping process [49]. This fork-related firing process is consistent with the observation that nearby origins tend to fire at similar times, which has been proposed to result from a different regulation of nearby and distant origins by Chk1 [33,35].

### 3.3. Modeling DNA Replication under Chk1 Inhibition and over Expression

To decipher the regulation of origin firing by Chk1, we examined if the MM5 model could also reproduce the replication program observed when the intra-S phase checkpoint was perturbed by the specific Chk1 inhibitor UCN-01 or by Chk1 over expression. We analyzed combed fibres from a replicated sample in the presence of UCN-01 (replicated fraction 22%) and in Chk1 over expression condition (replicated fraction 22%) that had spent the same interval of time in S phase as the control sample (global replicated fraction of 8% for UCN-01 and 46% in presence of Chk1 over expression). The MM5 model reproduced the experimental observations very well (Appendix A, $GoF_{global}\;=\;0.85$ for UCN-01 and $GoF_{global}\;=\;0.65$ for Chk1 over expression).

The two parameters *J* and θ were significantly higher in the UCN-01 treated sample than in the control samples with either the same harvesting time or a similar replicated fraction (22% and 19%, respectively) (Figure 6 and Appendix A). $P_{out}$ was higher in the UCN-01 treated sample than in the control samples with the same harvesting time but unchanged once comparing similar replicated fraction. In the same manner, *J*, and θ, were significantly lower in the Chk1 over expressed sample than in the control sample with the same incubation time (Figure 6b and Appendix A). However, $P_{out}$ and the other parameters were unchanged compared to control samples.

MM5 belongs to the general family of KJMA models that probabilistically describes the state of a nucleating and growing system [55]. In this framework, probabilities describing the nucleation are analogous to the probabilities of origin firing [56] and their values only depend on the parameters that describe the state of the system that in our case only the global fraction of replicated DNA. Hence, It seems natural that for two samples with the same replication fraction the values of probabilities $P_{out}$, $P_{in}$ and $P_{local}$ remain unchanged.

These results suggest that upon Chk1 inhibition (i) a fraction θ of the genome, where initiation probability is high, increases during S phase; (ii) the probability of origin firing is insensitive to Chk1 within this fraction ($P_{in}$ is unaltered) but is increased in the rest of the genome ($P_{out}$ is increased); (iii) the import/activation rate of the limiting factor, *J*, is increased, while the starting number of factors, $N_{0}$, is unaffected. As was expected, MM5 detected that Chk1 inhibition by UCN-01 increased origin firing [34,35,57,58,59,60]. However, upon Chk1 over expression (i) the fraction θ of the genome decreases, (ii) $P_{out}$ is insensitive to Chk1 over expression and (ii) the import/activation rate of the limiting factor, *J*, is decreased, while the starting number of factors, $N_{0}$, is unaffected. As was expected, MM5 detected that Chk1 over expression reduced the number of fired origins [35].

In conclusion, the level of Chk1 appears to regulate the kinetics of S phase progression (i) by limiting the genome fraction that escapes its inhibitory action, (ii) by down regulating the probability of origin firing outside this fraction [34,57,58,61] at the start of S phase, and (iii) by controlling the import/activation rate of limiting firing factors [34]. However, no significant differences in the strength of origin regulation by nearby forks ($P_{local}$) was observed after Chk1 inhibition or over expression, suggesting that this local action is not mediated by Chk1 [33,39].

## 4. Discussion

We explored several biologically plausible scenarios to understand the spatio-temporal organization of replication origin firing in *Xenopus* egg extracts. We used a quantitative approach to objectively discriminate which model best reproduced the genomic distributions of replication tracks as analyzed by DNA combing at different stages of S phase. We found that model MM5 with discrete potential origins best reproduced the experimental data with a minimal number of adjustable parameters. This model combines five assumptions [29,31,35,38,43,44,46,49,62,63]: (1) origin firing is stochastic, (2) the availability of a rate-limiting firing factor captures the essential dynamics of the complex network of molecular interactions required for origin firing, (3) the speed of replication forks is constant (4) origins fire in a domino-like fashion in the proximity of active forks [49,64]; (5) the probability of origin firing is heterogeneous along the genome [31,43].

We used MM5 to model DNA combing data from *Xenopus* egg extracts in presence or absence intra-S checkpoint inhibition and activation. In all conditions, this model was able to match the experimental data in a satisfactory manner. Furthermore, the inferred parameters values indicated that the global probability of origin firing and the rate of activation/import of the limiting firing factor (*J*) were increased after Chk1 inhibition by UCN-01 [34,59,65] and decreased after Chk1 over expression. Importantly, this model assumes a heterogeneous probability of origin firing and suggests that Chk1 exerts a global origin inhibitory action during unperturbed S phase [35] by following two possible mechanisms: (i) the first path corresponds to the regulation of the number of available replication limiting factors by Chk1 protein and (ii) the second path corresponds to the ability of Chk1 protein to reduce the capacity of potential origins to fire outside domains with high probability of origin firing. The strength of the second path decreases from the beginning of S phase to reach its minimal value after the first quarter of S phase. On the other hand, the constancy of the initial number of limiting factors $N_{0}$ in the presence or absence of UCN-01 or Chk1 over expression suggests that Chk1 does not actively control origins or the available number of replication limiting factors before S phase actually starts [36,66,67]. Interestingly, a better statistical match between the model and the data was obtained by assuming that the rate of DNA synthesis is variable downstream of replication forks. Indeed, the downstream of a replication forks the rate of DNA synthesis depends on the speed of replication fork and the frequency of firing of closeby potential replication origins [55]. Our analysis suggests that this variability cannot be mapped to a model with variable fork speed, but it is compatible with an increased probability of origin firing in the neighbourhood of an active replication fork. These observations indicate that MM5 can deliver a reliable, minimally complex picture of origin firing regulation in *Xenopus* egg extracts.

### 4.1. The Global Inhibition of Origin Firing by Chk1

We previously showed that Chk1 is active and limits the firing of some potential origins in an unperturbed S phase [35]. Therefore, the earliest origins must be immune to Chk1 inhibition while later potential origins are strongly inhibited. The comparisons among the modelling of Chk1 inhibition, over expression and of unperturbed S phase data suggests that (i) the probability of origin firing is reduced by active Chk1 in a fraction 1−θ of the genome, (ii) in this Chk1-sensitive fraction the probability of origin firing increases as S phase progresses and (iii) the probability of origin firing is unaffected by Chk1 inhibition within the Chk1-immune, θ fraction of the genome. Therefore, this model supports the idea that at the start of S phase, some origins fire unimpeded by Chk1, whereas others remain silent. The latter only becomes progressively relieved from Chk1 inhibition as S phase progresses. Indeed, recent works in cultured mammalian cells [68], *Drosophila* [60] and *Xenopus* [69] showed that in unperturbed S phase the global origin firing inhibitory effect (by Chk1 and Rif1) is reduced as S phase progresses. Interestingly, a recent study in unperturbed yeast cells suggests that dNTPs are limiting at the entry into S phase, so that, similar to *Xenopus* [70], the firing of the earliest origins creates a replication stress that activates the Rad53 checkpoint which prevents further origin firing. Rad53 activation also stimulates dNTP synthesis, which in turn down regulates the checkpoint and allows later origin firing [36]. However, it remains uncertain if this feed-back loop does also exist in *Xenopus* egg extracts which contain an abundant pool of dNTPs.

A key mechanism of our model is the enhancement of origin firing close to active forks. The necessity to introduce this mechanism supports the idea that the rate of DNA synthesis depends on the S-phase time and position of replication forks. During our modelling based on statistical ground we showed that the domino-like view of DNA replication progression [49,64] better describes the measured quantities from combed DNA molecules than the hypothesis of variable fork velocity [71]. It was previously shown in *Xenopus* egg extracts that the probability of origin firing could depend on the distance between left and right approaching forks [30]. While this could in principle reflect an origin firing exclusion zone ahead of forks [23,49], our model did not allow for a negative $P_{local}$ but the fact that the discrete distribution of potential replication origins better describes the experimental data than the continuous distribution confirm the necessity of the existence of origin firing exclusion zones between two converging replication forks. Other proposed mechanisms for origin clustering include the relief of Chk1 inhibition ahead of active forks by checkpoint recovery kinase polo like kinase 1 (Plk1) [35,39]. However, we find that the range, *d*, and the strength, $P_{local}$, of origin stimulation by nearby forks, were both insensitive to checkpoint inhibition or activation (Figure 6a,b). Other potential mechanisms such as propagation of a supercoiling wave ahead of forks may better explain this insensitivity to Chk1 inhibition [72].

### 4.2. Heterogeneous Probability of Origin Firing

In this model, the origin firing process in *Xenopus* egg extracts is not reliably described by a mean-field approximation. In other words, the probability of origin firing is heterogeneous along the genome. Based on this hypothesis, one important outcome of our study is that the genome can be segmented into domains where origin firing probability is either high and immune to Chk1 inhibition or low and subjected to a tight Chk1 control that attenuates as S phase progresses. This picture challenges the common view that the embryonic *Xenopus* in vitro system would lack the temporal regulation by the intra-S checkpoint at the level of large chromatin domains in contrast to findings in somatic vertebrate cells where Chk1 controls cluster or replication foci activation [61]. However, observations of replicating nuclei in *Xenopus* system have shown that early replication foci are conserved in successive replication cycles, supporting the heterogeneous domain hypothesis [32]. Furthermore, we found that the fraction of the genome covered by these domains increases and that the inhibitory action of Chk1 decreases over time during an unperturbed S phase (Figure 5 and Figure 6), consistent with the idea that as S phase progresses more regions of the genome evade the checkpoint inhibition of origins. By comparing samples that have spent the same time interval in S phase or that have reached the same replicated fraction in the absence and presence of UCN-01 (Figure 6a) or have spent the same time interval in S phase in a Chk1 over-expressed condition (Figure 6b), we noticed that the probability of origin firing in the Chk1-immune domains ($P_{in}$) did not change upon Chk1 inhibition or over expression. This further suggests that these domains actually escape the regulation of origin firing by Chk1 that rules the rest of the genome. It is an interesting observation that in Chk1-immune regions where the probability of origin firing is high, the temporal difference between two firing events would be smaller than in other regions of the genome. This leads to an observed synchrony of origin firing and therefore to an effective observed clustering of replication eyes on a single DNA fibre.

### 4.3. How the Model Is Applicable to Other Systems?

Recent studies of the spatio-temporal pattern of DNA replication in other metazoans, namely human and mouse, generated either replication timing profile for each chromosomes [73,74] or mapped the genome wide replication using optical mapping [75]. To generate a signal comparable to those obtained by these two technics, we modified the MM5 model by imposing that some regions with size $L_{i}$ of a fictitious L=200 Mbp chromosome have a probability of origin firing $P_{in}$, with the constraint that $\theta =\frac{\sum_{i=1}^{N}L_{i}}{L}$ with N=398 the number of regions and $L_{i}$ is randomly chosen for each region in the interval $\left[0,3\right]$ Mbp. We performed 100 times the complete chromosome replication simulation. The 100 independent simulations were pooled. The fraction of abundance for each position was calculated (Figure 7a).

Similar to human and mouse cell lines our in silico replication timing pattern can be segmented into plateaus of early and late constant timing regions (CTRs) separated by timing transition regions (TTRs) [74] (Figure 7b). Regions with high probability of firing ($P_{in}$) fire earlier than regions with lower probability of firing ($P_{out}$), confirming that the difference in early and late firing regions is induced by the relative difference in their intrinsic probabilities of origin firing and the firing event of an origin can extend over the whole S phase [75]. In our simulations early firing origins are excluded from the TTRs and the density of fired origin is smaller in this region than in CTRs. This is compatible with recent observations in human and mouse cell lines [73]. While MM5 implies that TTRs are constituted of fork induced initiations that fire very close to each others, they have not been detected experimentally by methods with a genomic resolution higher than 15 kb [73,75]. Finally, this model produces in early and late CTRs sites of origin firing randomly distributed and replicating very rapidly that induces an apparent clustering of origins, this phenomena is observed experimentally in human and mouse cell lines [73,75].

The above characteristics of origin firing process shared by the MM5 model and human and mouse cell line suggests that the spatio-temporal pattern of DNA replication in these organism can be analysed quantitatively in the framework defined by MM5. Interestingly, MM5 implies that, at the start of S phase the firing of some origins is immune to the inhibitory action of intra-S phase checkpoints as observed in human cell lines [75,76].

## 5. Conclusions

All together the results of our modelling approach and the existing literature suggest that in the *Xenopus* system the position of early replicating, Chk1-immune domains is conserved in individual nucleus. However, there is no experimental or numerical evidence that the positions of these domains are conserved in a population of nuclei. Assuming that the position of these domains changes randomly from one nucleus to another would result in a flat mean replication timing pattern and involves that each nucleus has its specific replication regulation process. While we cannot reject such a hypothesis objectively, the recent report of a structured replication timing program in zebrafish early embryos [54] encourages us also to assume the hypothesis that in *Xenopus* early embryos the position of early replication domains are conserved from one nucleus to another leading not to a flat but structured mean replication timing pattern similar to other eukaryotic systems [6,8,51]. The generality of assumptions and conclusions of our model suggest that it can be used to analyze the dynamics of S phase and its regulation by the intra-S phase checkpoint in other organisms.

## Figures and Tables

**Figure 1 genes-12-01224-f001:**
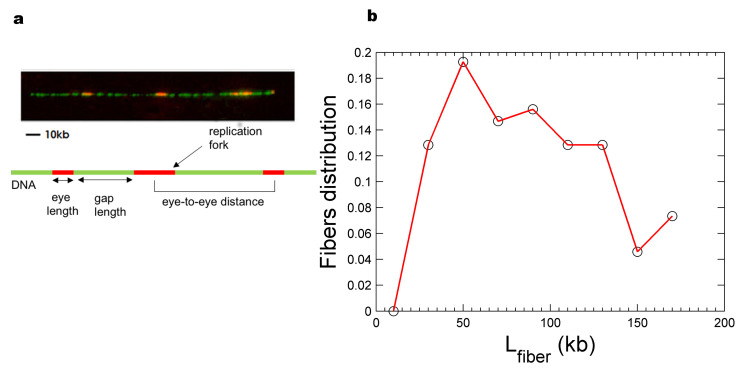
Characteristics of combed DNA molecules. (**a**). Example of combed DNA molecule. The top panel is a fluorescence microscopy of a representative, stretched DNA fiber (green) containing replication eyes (red). The bottom panel is a schematic illustration of measured parameters in replication studies using DNA combing. (**b**). Molecular length distribution (global replicated fraction of 8%) of combed DNA fibre. The black open circles are the experimentally measured and the red curve is the simulated cut molecular length distributions, respectively.

**Figure 2 genes-12-01224-f002:**
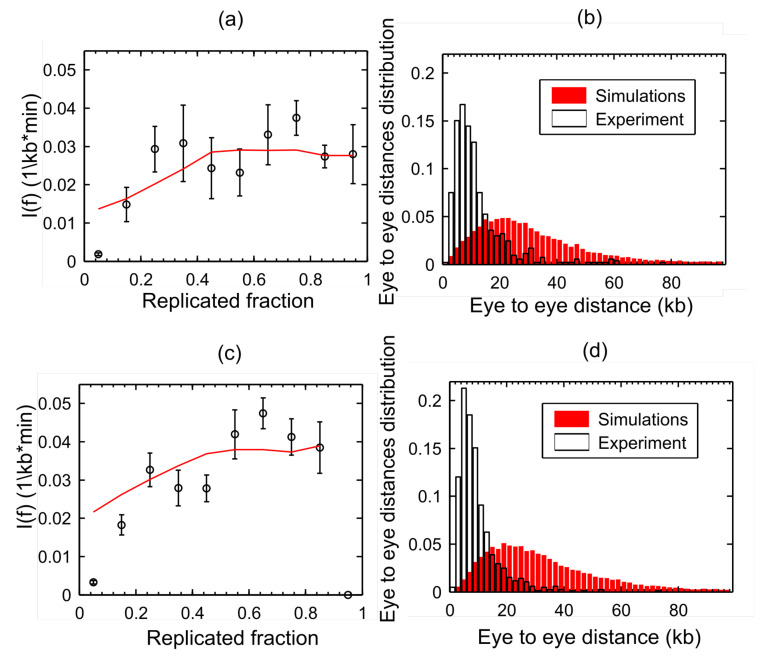
Chk1 does not control origin clustering. The black symbols are experimental data and the red curves are simulations. (**a**,**c**) Fitting of $I\left(f\right)$ data extracted from raw data published in [35] as described in material and methods for control and Chk1 inhibition experiments, respectively. The discrepancy in values between the extracted data and those published in [35] are due to difference in thresholding and the lack of smoothing of the extracted data in this work. (**b**,**d**) Discrepancy between experimental and simulated distributions of eye-to-eye distances in control and Chk1 inhibition experiments, respectively.

**Figure 3 genes-12-01224-f003:**
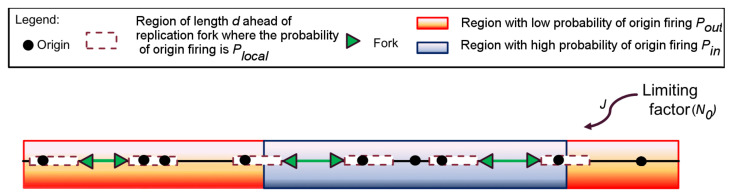
Schematic representation of MM5. Potential replication origins located in a fraction θ of the genome (not necessary contiguous) have a probability of firing $P_{in}$ higher than probability of firing $P_{out}$ of potential origins located in the complementary genome fraction 1−θ. The firing of a potential origins requires its encounter with limiting factors which number $N\left(t\right)=N_{0}+Jt$ increases as S phase progresses. Potential origins fire with a probability $P_{local}$ over a distance *d* ahead of a replication fork.

**Figure 4 genes-12-01224-f004:**
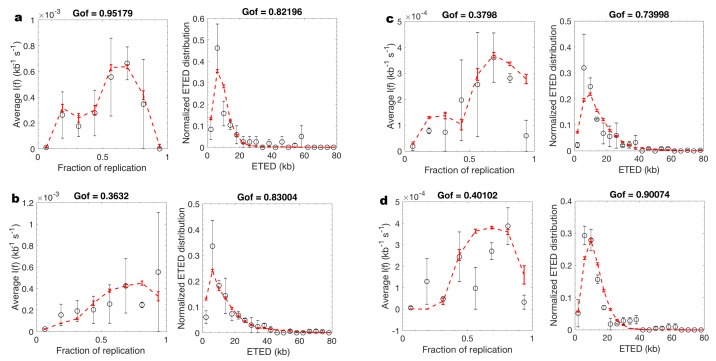
MM5 captures the essential processes necessary to model the regulation of DNA replication by Chk1. (**a**,**b**). Unchallenged (8% global replication fraction) and Chk1 inhibited samples (22% global replication fraction) corresponding to the same experiment and harvested at same time. (**c**,**d**). Unchallenged (46% global replication fraction) and Chk1 over expressed samples (22% global replication fraction) corresponding to the same experiment and harvested at same time. The black open circles are experimental data and the dashed red lines are the fit obtained by MM5 model.

**Figure 5 genes-12-01224-f005:**
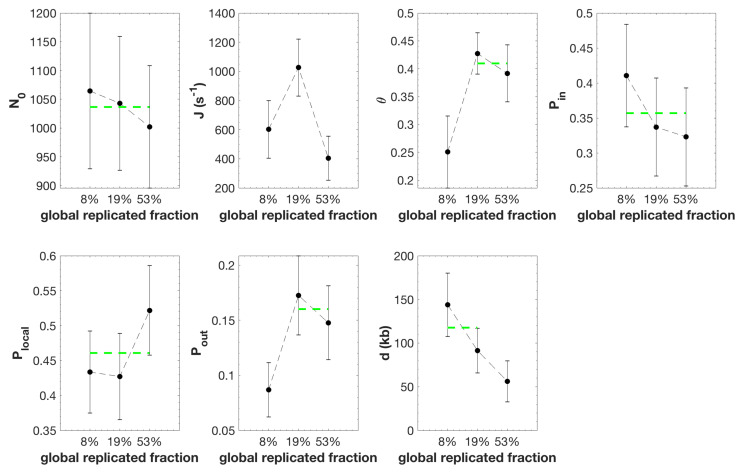
Inferred model parameters by fitting unchallenged S phase data as global replicated fraction increases. The black circles are the averaged value of the parameter over 100 independent fitting processes and the error bars are standard-deviations. The green dashed line is the mean value among consecutive parameters which differences are not statistically significant (Appendix A).

**Figure 6 genes-12-01224-f006:**
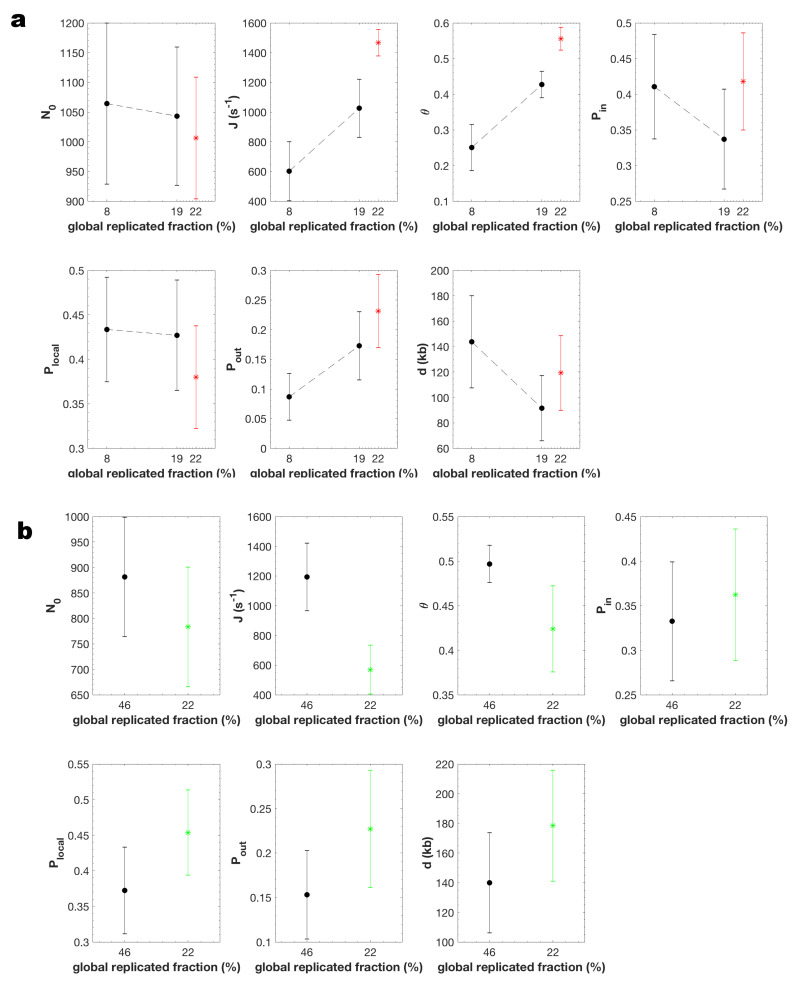
*J*, θ, and the $P_{out}$ are the only parameters that change when comparing unchallenged, (**a**) Chk1 inhibited and (**b**) Chk1 over-expressed S phase The black circle is the averaged value of the parameter over 100 independent fitting processes of unchallenged S phase and the error bars are standard-deviations. The red star (**a**) is the averaged value of the parameter over 100 independent fitting processes of Chk1 inhibited sample and the error bars represent the standard-deviations. The green star (**b**) is the averaged value of the parameter over 100 independent fitting processes of Chk1 over-expressed sample and the error bars represent the standard-deviations.

**Figure 7 genes-12-01224-f007:**
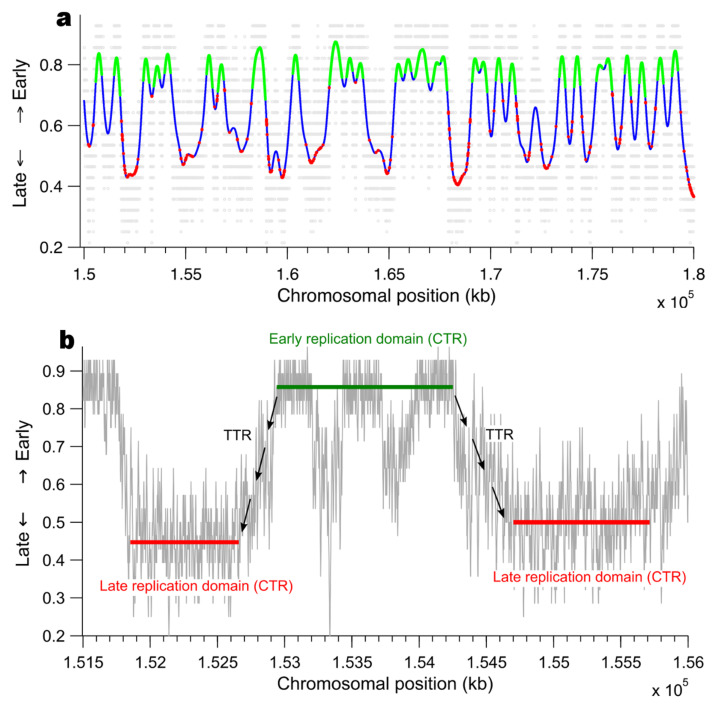
Replication timing profile generated by MM5. (**a**) The replication timing profile of a fictitious chromosome L=200 Mbp ($N_{0}=1231$, $J=287\;s^{-1}$, θ=0.46, $P_{in}=0.34$, $P_{local}=0.34$, $P_{out}=0.01$, d=138 kb). The grey dots are raw data and the blue line is the loess-smoothed curve. The green filled circles represent early firing origins and the red filled circles the late ones. (**b**) Replication timing profile for chromosomal positions $\left[1.515,1.56\right]\times 10^{5}$ kb. The profile can be segmented into plateaus of early and late constant timing regions separated by timing transition regions.

**Table 1 genes-12-01224-t001:** Lower and upper bounds of adjustable variables.

Variable	Lower Bound	Upper Bound	Significance
$N_{0}$	1	2000	Initial number of limiting-factor
$J\;(s^{-1})$	0	4000	Rate at which the number of limiting-factor increases
$P_{out}$	0	1	Probability of origin firing in the 1−θ fraction
$P_{in}$	0	1	Probability of origin firing in the θ fraction
$P_{local}$	0	1	Probability of origin firing ahead of an active replication fork over a distance *d*
θ	0	1	Fraction of genome where the probability of origin firing is $P_{in}$
d(kb)	0	1000	Distance over which a fork acts on the probability of origin firing

## Data Availability

The datasets supporting the conclusions of this article are from reference [35].

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
