# Peer review of "Organization of DNA Replication Origin Firing in Xenopus Egg Extracts: The Role of Intra-S Checkpoint"

_genes, 2021, doi:10.3390/genes12081224_

Round 1

Reviewer 1 Report

In this manuscript, Ciardo and colleagues present an improved mathematical model to describe experimental observations generated from profiling the in vitro DNA replication program in xenopus by prior DNA combing experiments.  This work builds on earlier efforts from the group and the models do show some improvement especially in the accurate prediction of interorigin distance (eye to eye).  They also consider the effect of perturbing the intra-s-phase checkpoint by modeling data from Chk1 inhibited or activated experiments.   Technically, the work is sound, albeit perhaps incremental in the context of their previous modeling efforts.  The model (mm5) is based on 5 assumptions i) origin firing is random, ii) avaliability of a rate limiting activating factor, iii) constant fork speed, iv) origins activate in a cascade relative to the fork and vi) the probability of origin fire is heterogenous.  The authors also model replication parameters in the context of chk1 inhibition and overexpression -- and attempt to place their findings in the context of the literature and what is known about the impact of chk1 inhibition (i.e. additional origins firing).  As an experimentalist, my first thought is how applicable is the model to other systems? Does the model developed from xenopus extracts also accurately predict the replication characteristics observed in recent genome wide combing experiments across the human genome (Wang et al, Mol Cell 2021)?  At least some discussion is warranted comparing the xenpus and human data as there did not appear to be any evidence for clustering of initiation events in the human data.   

Reviewer 2 Report

In this paper the authors analyze the organization of DNA replication in Xenopus egg extracts. They compare experimental results with simulations. Based on their results, the authors conclude that the genome has high and low probability of origin firing regions, high probability regions can escape the intra S checkpoint and the importance of DNA replication rates in different regions of the genome.

Overall, the results are interesting for researchers in the field and the data are convincing and the quality of the data is good. I have some suggestions for improvements that are listed below.

Specific points:

Even if it was already published elsewhere, the authors should describe the conditions of Chk1 inhibition by UCN-01 (time of treatment, concentration, etc).

The authors should describe the conditions of Chk1 overexpression (designed strategy, time of treatment, etc).

In figure 5 standard deviations are very big, can the authors conclude that the results are statistically significant?

Usually, inhibition and overepression experiments give opposite results. Tha authors shoud comment why some examined parameters are insensitive to Chk1 overexpression (pag 10, lines 299-302).

The authors conclude that “this local action is not mediated by Chk1” (pag 10, line 308) do they have any suggestion of proteins that might be involved?

The results are very interesting, but I am wondering if the conclusions are universal or limited to the used system. Are they replicable in other DNA replication models?
